# Maternal Immune Cell and Cytokine Profiles to Predict Cardiovascular Risk Six Months after Preeclampsia

**DOI:** 10.3390/jcm11144185

**Published:** 2022-07-19

**Authors:** Malia S. Q. Murphy, Samantha J. Benton, Brian Cox, Kara Nerenberg, Scott McComb, Lakshmi Krishnan, Risini D. Weeratna, Jean-François Paré, Alysha L. J. Dingwall-Harvey, Shannon A. Bainbridge, Andrée Gruslin, Laura M. Gaudet

**Affiliations:** 1Clinical Epidemiology Program, Ottawa Hospital Research Institute, Ottawa, ON K1H 8L6, Canada; malmurphy@ohri.ca (M.S.Q.M.); alyharvey@ohri.ca (A.L.J.D.-H.); 2Department of Obstetrics and Gynaecology, Faculty of Health Sciences, Queen’s University, Kingston, ON K7L 3N6, Canada; samanthabenton@cunet.carleton.ca; 3Department of Physiology, Faculty of Medicine, University of Toronto, Toronto, ON M5S 1A8, Canada; b.cox@utoronto.ca; 4Libin Cardiovascular Institute of Alberta, University of Calgary, Calgary, AB T2N 4N1, Canada; kara.nerenberg@ucalgary.ca; 5Departments of Medicine, Obstetrics & Gynecology and Community Health Sciences, University of Calgary, Calgary, AB T2N 4N1, Canada; 6Human Health Therapeutics, National Research Council of Canada, Ottawa, ON K1A 0R6, Canada; scott.mccomb@nrc-cnrc.gc.ca (S.M.); lakshmi.krishnan@nrc-cnrc.gc.ca (L.K.); risini.weeratna@nrc-cnrc.gc.ca (R.D.W.); 7Department of Biochemistry, Microbiology and Immunology, Faculty of Medicine, University of Ottawa, Ottawa, ON K1H 8M5, Canada; 8Department of Biomedical and Molecular Sciences, Queen’s University, Kingston, ON K7L 3N6, Canada; jfp3@queensu.ca; 9Interdisciplinary School of Health Sciences, Faculty of Health Sciences, University of Ottawa, Ottawa, ON K1N 7K4, Canada; shannon.bainbridge@uottawa.ca; 10Department of Cellular and Molecular Medicine, Faculty of Medicine, University of Ottawa, Ottawa, ON K1H 8M5, Canada; 11Department of Obstetrics, Gynecology and Newborn Care, The Ottawa Hospital, Ottawa, ON K1H 8L6, Canada; agruslin@toh.ca; 12Department of Obstetrics and Gynecology, Faculty of Medicine, University of Ottawa, Ottawa, ON K1H 8L6, Canada; 13Department of Obstetrics and Gynecology, Kingston Health Sciences Centre, Kingston, ON K7L 2V7, Canada; 14School of Epidemiology and Public Health, University of Ottawa, Ottawa, ON K1G 5Z3, Canada

**Keywords:** preeclampsia, cardiovascular risk, cardiovascular disease, immune cells, cytokines, biomarkers

## Abstract

Women who develop preeclampsia (PE) are at high risk for cardiovascular disease (CVD). Early identification of women with PE who may benefit the most from early cardiovascular risk screening and interventions remains challenging. Our objective was to assess whether cytokine and immune cell profiles after PE are helpful in distinguishing women at low and high CVD risk at 6-months postpartum. Individuals who developed PE were followed for immune cell phenotyping and plasma cytokine quantification at delivery, at 3-months, and at 6-months postpartum. Lifetime CVD risk was assessed at 6-months postpartum, and the immune cell and cytokine profiles were compared between risk groups at each time point. Among 31 participants, 18 (58.1%) exhibited high CVD-risk profiles at 6-months postpartum. The proportion of circulating NK-cells was significantly lower in high-risk participants at delivery (*p* = 0.04). At 3-months postpartum, high-risk participants exhibited a lower proportion of FoxP3^+^ regulatory T-cells (*p* = 0.01), a greater proportion of CD8^+^ T cells (*p* = 0.02) and a lower CD4^+^:CD8^+^ ratio (*p* = 0.02). There were no differences in immune cell populations at 6-months postpartum. There were no differences in plasma cytokines levels between risk groups at any time point. Subtle differences in immune cell profiles may help distinguish individuals at low and high CVD risk in the early postpartum period and warrants further investigation.

## 1. Introduction

Preeclampsia (PE) is a serious obstetrical complication that affects 3–7% of pregnancies worldwide [1,2]. Diagnosis generally occurs when hypertension develops concurrently with dysfunction of one or more organ systems, commonly the central nervous, liver and kidney systems [3]. The underlying causes of PE are poorly understood; however, the diversity of symptoms and histopathology associated with PE suggests that this disorder is a heterogeneous disease, arising via multiple mechanistic pathways [4,5,6].

PE is an established risk factor for metabolic, cardiovascular and cerebrovascular disease. Individuals who develop PE are three to four times more likely to develop chronic hypertension and twice as likely to develop heart disease or stroke after pregnancy [7,8]. The timing of the onset in pregnancy and the severity of symptomology appear to be meaningful, with earlier and more severe manifestations of PE being associated with greater risk of future disease [9]. Whether PE lies along a causal pathway that predisposes affected individuals to CVD or whether common cardiovascular risk factors contribute to the manifestation of both PE and CVD remains unclear. In any case, PE and other hypertensive disorders of pregnancy are recognized as a risk indicator by many leading cardiovascular and obstetrical organizations, who now recommend that obstetrical history be included as part of a woman’s cardiovascular and cerebrovascular risk evaluation [10,11,12].

Maternal endothelial dysfunction characteristic of PE is accompanied by systemic activation of the immune system and increased production of inflammatory cytokines [13]. The extent to which immune system activation persists post-delivery, however, has not been well characterized. Cytokines and immune cells are mediators of endothelial dysfunction, and their persistent disruption may contribute to the predisposition toward later disease in this unique population. Persistent inflammation often accompanies obesity, insulin resistance and other cardiovascular risk factors, and the link between immune cell and cytokine profiles and CVD risk in individuals who develop PE warrants further exploration. Our objectives were to assess changes in cytokine and immune cell profiles over the first 6 months postpartum following PE and to determine whether there are differences in cytokine and immune cell profiles at the time of delivery that may be helpful in identifying individuals at high risk of CVD for whom postpartum cardiovascular risk counselling and intervention would be of greatest value.

## 2. Materials and Methods

### 2.1. Study Design and Setting

This prospective cohort study was conducted at The Ottawa Hospital in Ottawa, Canada. Participants were recruited from antenatal clinics, triage units or during hospitalization for delivery. The study consisted of three visits: an initial visit at the time of recruitment, and follow-up visits at three and six months postpartum.

### 2.2. Study Participants

Patients with PE who were ≥18 years of age, carrying a live, singleton fetus and who were able to provide written informed consent in either English or French were eligible for this study. Those with pre-pregnancy kidney disease, >100 mg urine creatinine or proteinuria (>300 mg in a 24 h urine, or >30 mg/dL protein:creatinine ratio by spot urine, or >28 mg/dL microalbumin:creatinine ratio by spot urine) were excluded, as were individuals with pre-existing or gestational diabetes, pre-existing CVD and major maternal or fetal complications other than PE.

PE was defined according to diagnostic criteria from the Society of Obstetricians and Gynaecologists of Canada [3], which include hypertension (blood pressure ≥140/90 mmHg, on at least two occasions > 15 min apart after 20 weeks’ gestation) with new proteinuria (≥0.3 g/day by 24 h urine collection, ≥30 mg/mmol by protein:creatinine ratio or ≥1+ by urinary dipstick) one osr more adverse conditions (e.g., headache/visual symptoms, chest pain/dyspnea, nausea or vomiting, right upper quadrant pain, elevated white blood cell count) or one or more severe complications (e.g., eclampsia, uncontrolled severe hypertension, platelet count <50 × 10^9^/L, acute kidney injury).

### 2.3. Data Collection

Baseline data were collected through participant interviews and retrospective reviews of antenatal records. Data included socio-demographic characteristics (maternal age, education, employment status), health behaviours (smoking, alcohol consumption, physical activity before and during pregnancy), biophysical profiles (height, pre-pregnancy weight), familial medical history (coronary artery disease, cerebrovascular disease, type 1 or 2 diabetes, hypertension, malignancy, obesity, hypertensive disorders of pregnancy), maternal obstetrical history (parity, complications in previous pregnancies) and delivery characteristics (last recorded blood pressure, maternal weight, gestational age, mode of delivery, infant birthweight).

Participant physical profiles were collected at six months postpartum for the calculation of lifetime cardiovascular risk [14,15,16]. Physical measurements included maternal weight, blood pressure (an average of five readings using a BpTRU^TM^ Blood Pressure Monitor; VSM MedTech, Coquitlam, BC, Canada), waist circumference, hip circumference and smoking status. Physical activity at 6-months postpartum was also collected. Lifetime cardiovascular risk was calculated based on the status of seven risk factors: total cholesterol (<4.65 mmol/L, optimal; 4.65–5.15 mmol/L, not optimal; 5.16–6.19 mmol/L, elevated; >6.20 mmol/L major), systolic blood pressure or current use of antihypertensive medication (<120 mmHg, optimal; 120–139 mmHg, not optimal; 140–159 mmHg, elevated; ≥160 mmHg or current use of antihypertensive medication, major), diastolic blood pressure or current use of antihypertensive medication (<80 mmHg, optimal; 80–89 mmHg, not optimal; 90–99 mmHg, elevated; ≥100 mmHg or current use of antihypertensive medication, major), fasting glucose or previous diagnosis of type 1 or 2 diabetes (≤6.88 mmol/L, optimal, >6.88 mmol/L or diabetic, major) and smoking status (no, optimal; yes, major). Lifetime cardiovascular risk scores were classified as 8% (all risk factors are optimal), 27% (≥1 risk factor is not optimal), 39% (≥1 risk factor is elevated OR 1 risk factor is major) and 50% (≥2 risk factors are major), then simplified as low (<39%) or high (≥39%) [14,15,16]. Baseline, pre-pregnancy maternal cardiovascular risk profiles were not available.

### 2.4. Sample Collection and Processing

Blood samples were collected within 72 h of delivery and at three and six months postpartum. At each collection, 10 mL of maternal blood was collected by venipuncture into EDTA Vacutainers, and plasma was isolated by centrifugation (3000 rpm, room temperature, 10 min). Aliquots were stored at −80 °C for subsequent cytokine analysis in batch assays. Another 10 mL of maternal blood was collected into lithium heparin vacutainers, and peripheral blood mononuclear cells (PBMC) were isolated by density gradient centrifugation using Ficoll^®^ Pacque Plus (GE Healthcare Life Sciences, Mississauga, ON, Canada). The PBMCs were collected and washed with 1X PBS twice. Pellets were re-suspended in 10% DMSO/FBS and stored at −150 °C until immune cell phenotyping in batch assay. Additional maternal blood was collected for C-reactive protein, fasting glucose, HbA1C, LDL, HDL, triglycerides and total cholesterol quantification, and urine was collected for albumin:creatinine ratio by the hospital laboratory.

### 2.5. Cytokine Quantification

Plasma cytokines were measured using the Bio-Plex Pro^TM^ Human Cytokine 17-plex Assay kit (Bio-Rad; Hercules, CA, USA) using the Bio-Plex 200 system (Bio-Rad; Hercules, CA, USA). The panel included: G-CSF, GM-CSF, IFN-γ, IL-1β, IL-2, IL-4, IL-5, IL-6, IL-7, IL-8, IL-10, IL-12 (p70), IL-13, IL-17A, MCP-1, MIP-1β and TNF-α. Plates were read on a Bio-Plex Luminex 200 System (Bio-Rad).

### 2.6. Immunophenotyping

Immunophenotyping was conducted by multiplex 16-color flow cytometry on an LSR-Fortessa^®^ analyzer (BD, 5-laser, 18 colour) at the National Research Council laboratory in Ottawa, Canada using a fluorescence-activated cell staining (FACS) protocol developed to examine a broad array of immune cell types in cryopreserved PBMC samples. For a full list of antibodies used see Appendix A. Thawed PBMC were assessed for viability using a Cellometer Auto 2000 (Nexcelcom, Lawrence, MA, USA) automated cell counter. PBMC were then suspended in a single cell suspension normalized up to 3 × 10^6^ live cells/100 mL of FACS buffer. Cells were incubated at 4 °C with surface staining antibodies (all markers excluding FoxP3) and washed with PBS. Cells were then incubated with dead cell stain (LIVE/DEAD^®^ Fixable Aqua fluorescent reactive dye) for 30 min at 4 °C. Cells were washed with FACS buffer, re-suspended in permeabilisation buffer and incubated for 30 min at room temperature. Cells were then washed and re-suspended in 100 µL of FACS buffer. FoxP3 antibody was added to the solution, and cells were incubated for 30 min at room temperature. Cells were washed with FACS buffer and re-suspended in 300 µL of buffer for analysis. Panel testing and inter- and intra-assay consistency was assessed (see Appendix A).

Post-acquisition, the percentages of 18 different immune cell populations (including B cells, T cells, NK cells, monocytes and subsets thereof) were determined using FlowJo software (Tree Star Inc., Ashland, OR, USA) (Table 1). FACS data were scored as a percentage of the parent or grandparent population. Specific gating schema employed are provided in Appendix A.

### 2.7. Statistical Analysis

Data were analysed using Prism 4.0 (GraphPad, San Diego, CA, USA). Descriptive data were summarized using frequencies (*n*) and percentages (%) for categorical data, and median and interquartile ranges (IQR) for continuous data. Cytokine and immune cell data were stratified by lifetime CVD risk status at six months postpartum (low risk versus high risk). The Wilcox rank sum test was used to assess statistical differences in percentage cell populations for FACS and cytokine data at each study time point. *p* < 0.05 was considered significant.

## 3. Results

A total of 47 individuals with PE were enrolled into the study, and 31 (66.0%) completed the 6-month study visit and cardiovascular risk screening. The data presented in this study are limited to the 31 participants and for whom lifetime cardiovascular risk scores were available.

### 3.1. Participant Characteristics

Participants had a median (IQR) age of 33 (30, 38) years and pre-pregnancy BMI of 27.4 (23.3, 34.0) kg/m^2^ (Table 2). The majority had a college or university education (93.5%) and had fulltime employment before delivery (77.4%). Less than 50% of participants were physically active (≥150 min/week moderate to vigorous physical activity) before or during pregnancy. The prevalence of smoking in pregnancy among participants was low (3.2%).

At 6-months postpartum, 11 (35.5%) and 14 (45.2%) participants had systolic and diastolic blood pressures above the optimal range, respectively, and an additional 5 participants were taking antihypertensive medication (Table 3). A total of 19 (61.3%) participants had total cholesterol levels above the optimal range. No participants had elevated fasting glucose levels. One participant reported smoking at the 6-months postpartum visit. Based on their systolic and diastolic blood pressures, fasting glucose, total cholesterol and smoking status, 18 (58.1%) participants were classified at high (≥39%) lifetime risk for CVD.

### 3.2. Immune Cell Profiles

Immune cell data were available from 15 (48.4%) participants at delivery, 16 (51.6%) at 3-months postpartum and 28 (90.3%) at 6-months postpartum. Immune cell data were stratified by lifetime CVD risk status assessed at 6-months postpartum.

Circulating T cells at the time of delivery did not correlate with CVD risk at 6-months postpartum (Figure 1). At 3-months postpartum, however, participants at high lifetime risk of CVD exhibited higher percentages of cytotoxic CD8^+^ T cells (*p* = 0.0204). Accordingly, CD4:CD8 T cell ratios, which have been associated with CVD risk, were lower in the high-risk CVD group at 3-months postpartum (*p* = 0.024). The proportion of regulatory T cells (CD4^+^ FoxP3^+^), known to modulate inflammation, was significantly decreased at 3-months postpartum in the high-risk CVD group compared to the low-risk CVD group (*p* = 0.014), but differences were resolved at 6-months postpartum (*p* = 0.192).

Other non-T cell lymphocytes were examined. The proportion of circulating NK cells was significantly decreased at the time of delivery in high-risk CVD participants compared to low-risk CVD participants (*p* = 0.0401), and not significantly elevated at 3-months postpartum (*p* = 0.081) (Figure 2). There were no statistical differences between groups at 6-months postpartum (*p* = 0.591). NK subsets were also examined, but no statistical differences between the study groups were observed across the time points.

Circulating B cells did not correlate with CVD risk at 6-months postpartum, and there were no variations in the number of monocytes (CD14^+^ CD11b^+^) or inflammation-associated loss of HLA-DR in the monocyte sub-population.

Tabular data for all immune cell populations analyzed by study time point and comparison group are provided in the Appendix A.

### 3.3. Cytokine Profiles

A total of 28 (90.3%) participants provided sufficient samples for cytokine profiling at delivery, 29 (93.5%) at 3-months postpartum and 27 (87.1%) at 6-months postpartum. There were no differences in maternal plasma IL-6, IL-8, MCP-1, MIP-1B and TNF-α between the CVD risk groups, across any of the study time points (Figure 3). All other cytokines (G-CSF, GM-CSF, IFN-γ, IL-1β, IL-2, IL-4, IL-5, IL-7, IL-10, IL-12 (p70), IL-13 andIL-17A) measured were below the assay limit of detection.

## 4. Discussion

In this study, 31 women who developed PE during pregnancy were followed from delivery to 6-months postpartum for immune cell and cytokine profiling and CVD risk screening. Using the presence of traditional cardiovascular risk factors at 6-months postpartum, 13 (41.9%) participants were assessed to be at high (≥39%) lifetime risk for CVD. Participants at high lifetime risk of CVD exhibited significant reductions in NK cells peri-delivery and notable changes in FoxP3^+^ and CD8^+^ T-cells at 3-months postpartum compared to participants with low lifetime risk of CVD. Pro-inflammatory phenotypes resolved by 6-months postpartum.

We sought to determine whether there were measurable differences in immune cell and cytokine profiles in the early postpartum period that may be useful for identifying women with particularly high risk for future CVD. A diagnosis of PE provides a window of opportunity for CVD prevention through optimization of modifiable risk factors including treatment of hypertension, diabetes and dyslipidemia. It is during pregnancy and in the first year postpartum that a woman is highly engaged with the health system [17]. However, not all women with a history of PE will go on to develop CVD, and clinicians lack simple tools to identify women at greatest risk for whom targeted follow-up may be of greatest benefit [18]. Biomarkers of inflammation are important independent predictors of cardiovascular events in the general population [19,20,21] yet are not validated in the postpartum obstetrical population.

Peri-delivery, we observed a reduction in NK cells among women who had a high lifetime risk for CVD at 6-months postpartum, though no statistical alterations in NK subsets were noted. Although not itself a definitive discriminator, in combination with other more traditional CVD risk factors, NK cells may be useful in identifying those at highest risk and warrant further investigation. Maternal endothelial dysfunction associated with PE is attributed to immune system dysregulation and overstimulation of the complement system, the severity of which is correlated with severity of the PE itself [22,23,24,25,26]. NK cells play a significant role in this process, and activation of decidual NK cells in PE are well described [27]. In contrast, there are conflicting data on whether peripheral NK cell subsets are altered in PE and their value for providing insight into placental development [27]. Although no other immune cell types showed significant divergence peri-delivery based on CVD profile, the profound inflammatory effects of PE could mask subtle underlying differences.

Data on the persistence of immunoregulatory alterations later into the postpartum period are also lacking. In a small cross-sectional study, Kieffer et al. reported persistent effects of PE on memory T cell populations in a small sample of women at least 6 months after delivery [28]. Women with a history of PE exhibited lower proportions of activated CD4^+^ memory T cells and CD8^+^ effector memory cells compared to women who had experienced a healthy pregnancy, suggesting a possible involvement of memory T cells in the recurrence risk of PE. The lower proportion of FoxP3^+^ regulatory T cells, combined with increases in the proportion of CD8^+^ T cells at 3-months postpartum among participants with high-risk CVD profiles in our study suggests perpetuation of a pro-inflammatory phenotype long after the clinical resolution of PE. FoxP3^+^ regulatory T cells are atheroprotective and critical in the maintenance of immune tolerance and regulation of the immune system [29,30]. Depletion of regulatory T cells results in alterations in lipid metabolism, hypercholesterolemia and doubling of atherosclerotic plaque size in animal models [31]. Reductions in circulating regulatory T cells are also evident in patients with CVD [32,33]. CD4 and CD8 T cell subtypes are implicated in PE, hypertension and atherosclerosis although the complex co-stimulatory and co-inhibitory pathwsays regulating T cell activation in these conditions remains incompletely understood [34,35]. In this study, immune cell differences peri-delivery and at 3-months postpartum were resolved by 6-months postpartum; so whether our observations are truly linked to CVD risk remains unclear.

Despite evident immune cell changes, we did not find differences in cytokine levels between postpartum PE women at low versus high cardiovascular risk. Our findings are similar to those of another study in which circulating cytokine levels and lifetime CVD risk were examined in a small cohort of PE (*n* = 35) and control (*n* = 28) women at 6-months postpartum [36]. The authors found no differences between PE and control participants for 14 cytokines including IL-6, IL-8, IL-10, TNF-α and INF-y, and there was no association between cytokine levels and postpartum CVD risk [36]. However, alterations in circulating cytokine levels after PE have been described elsewhere [37] and are supported by evidence of endothelial dysfunction and inflammatory differences [38,39] in the months to years after PE [40,41,42].

The strengths of this study include the prospective design with extensive data collection and characterization of immune cell and cytokine profiles. This was a small, single-centre evaluation, however, and low sample size leaves us underpowered to detect subtle differences between lows- and high-risk groups. Other than lower NK cell numbers, we did not find significant disruptions to immune cell and cytokine profiles peripartum as was expected. This may be a consequence of multiple factors including small sample size, sampling up to 72 h post-delivery and our inclusion of individuals with any PE diagnosis without distinguishing between different PE sub-types. As a result, our ability to infer on the predictive potential of circulating markers at the time of delivery for later CVD risk is limited. In the event that blood-based measures prove helpful in the prediction of premature cardiovascular disease in the postpartum population, their added benefit relative to the cost of incorporating immune cell and cytokine profiling into otherwise non-invasive methods for cardiovascular risk screening will need to be considered. Our study was also affected by loss-to-follow-up over the 6-month study period. A total of 47 participants consented to the study and provided blood samples at delivery, and only 66% completed cardiovascular risk screening at 6-months postpartum. Finally, we are unable to contextualize our findings relative to a control group with normal pregnancies or to participant pre-pregnancy cardiovascular risk status. While biomarkers of endothelial dysfunction including immune cells and cytokines are rarely measured before pregnancy, we cannot comment on whether pre-existing immune derangements in our participants may have contributed to our findings.

## 5. Conclusions

Although existing data are derived from studies limited by small sample size, together, the literature suggests that the impacts of PE on immune system regulation may persist after pregnancy when the signs and symptoms of the condition are expected to be resolved. Our findings suggest the same. Fewer NK cells at delivery and differences in FoxP3^+^ and CD8^+^ T-cell populations at 3-months postpartum suggest the perpetuation of pro-inflammatory phenotypes post-delivery among a subset of women who go on to develop high-risk CVD profiles. Longer-term and larger follow-up studies will be required to validate these findings. Ultimately, in the search for effective approaches to postpartum cardiovascular risk screening, immunological profiling alone is unlikely to be the solution. Instead, prediction models that integrate traditional risk factor data, placental pathology findings and biomarker data may be required. Further investigation in this area will inform clinical postpartum cardiovascular risk factor screening and intervention strategies [19] by helping to identify individuals at highest risk who are most likely to benefit from postpartum interventions.

## Figures and Tables

**Figure 1 jcm-11-04185-f001:**
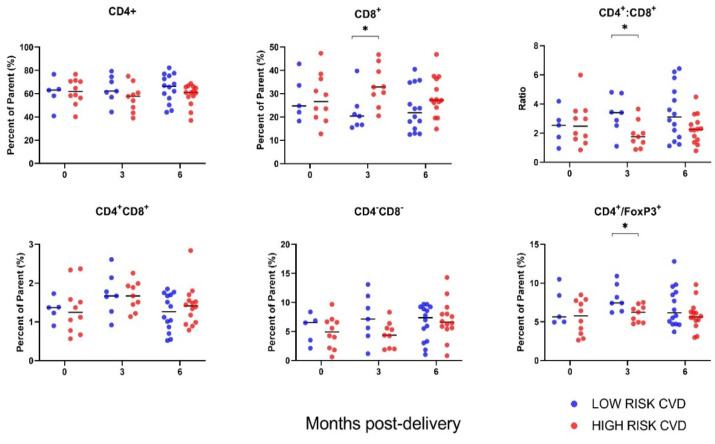
Maternal T-cell profiles by CVD risk. Immune cell data are stratified by participant lifetime CVD risk status at 6-months postpartum (low versus high). Results are reported as percent of the parent cell population (live/size/CD3^+^ CD45^+^ cells for all except for FoxP3^+^ T cells, for which the parent cell population is live live/size/CD3^+^ CD45^+^ /CD4^+^ T cells). Sample sizes at each time point: *n* = 15 at delivery, *n* = 16 at 3-months postpartum and *n* = 28 at 6-months postpartum. * *p* < 0.05.

**Figure 2 jcm-11-04185-f002:**
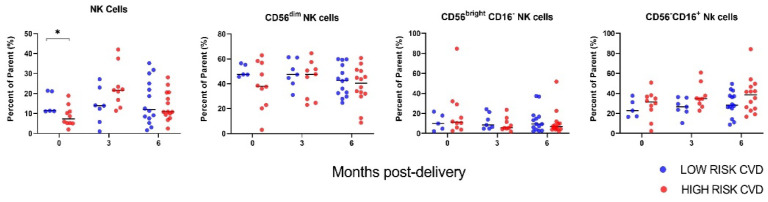
Maternal NK-cell profiles by CVD risk. Immune cell data are stratified by participant lifetime CVD risk status at 6-months postpartum (low versus high). Results are reported as percent of the parent cell population (live/size/CD3^−^ CD45^+^ cells for all). Sample sizes at each time point: *n* = 15 at delivery, *n* = 16 at 3-months postpartum and *n* = 28 at 6-months postpartum. * *p* < 0.05.

**Figure 3 jcm-11-04185-f003:**
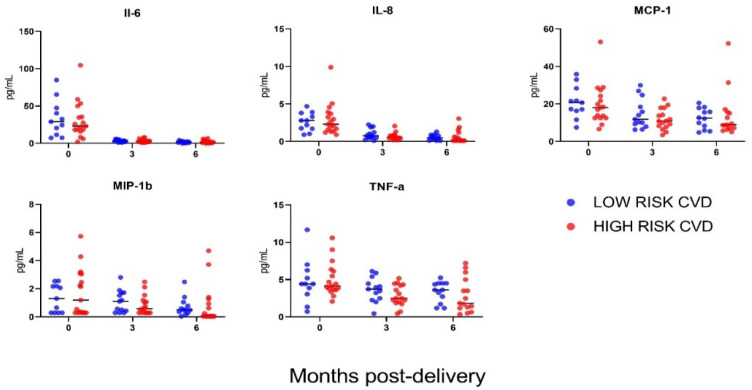
Maternal serum cytokine profiles. Cytokine data are stratified by participant lifetime CVD risk status at 6-months postpartum (low versus high). Sample sizes at each time point: *n* = 28 at delivery, *n* = 29 at 3-months postpartum and *n* = 27 at 6-months postpartum.

**Table 1 jcm-11-04185-t001:** Evaluated immune cells phenotypes.

Immune Cell Families	Phenotypes and Parent Populations
T Cells	CD3^+^ CD45^+^ T cells
	└ CD4^+^ T cells
	└ FoxP3^+^ regulatory T cells
	└ CD8^+^ T cells
	└ CD4^+^ CD8^+^ (double positive) T cells
	└ CD4^−^ CD8^−^ (double negative) T cells
	└ CD25^+^ activated T cells
	└ CD69^+^ activated T cells
Non T Cell lymphocytes and Myeloid cells	CD3^−^CD45^+^
	└ CD3^−^ CD45^+^ CD19^+^ B cells
	└ HLA DR^+^ B cells
	└ CD3^−^ CD45^+^ NK cells
	└ CD56^bright^ CD16^−^ NK cells
	└ CD56^dim^ NK cells
	└ CD56^neg^ CD16^+^ NK cells
	└ CD14^+^ Myeloid cells
	└ CD11b^+^ Monocytes
	└ HLA-DR^+^ Monocytes

**Table 2 jcm-11-04185-t002:** Baseline characteristics of study participants.

Characteristic	Total Cohort, *n* = 31
Maternal Age, years, median (IQR)	33 (30, 38)
College/University education, *n*(%)	29 (93.5)
Fulltime employment, *n*(%)	24 (77.4)
Current smoker, *n*(%)	1 (3.2)
Pre-pregnancy alcohol consumption, *n*(%) ^b,c^	15 (48.4)
Physically active ^a,d^, *n*(%)	
Before pregnancy	12 (38.7)
During pregnancy	9 (29.0)
Pre-pregnancy BMI, kg/m^2^, median (IQR)	27.4 (23.3, 34.0)
Underweight (<18.5 kg/m^2^), *n*(%)	1 (3.2)
Healthy (18.5–24.9 kg/m^2^), *n*(%)	11 (35.5)
Overweight (25–29.9 kg/m^2^), *n*(%)	7 (22.6)
Obese (≥30 kg/m^2^), *n*(%)	12 (38.7)
Maternal history of complications in pregnancy, *n*(%)	
Gestational hypertension	7 (22.6)
Preeclampsia	6 (19.4)
Intrauterine growth restricted fetus	1 (3.2)
Placental abruption	1 (3.2)
Preterm birth < 34 weeks	3 (9.7)
Known family medical history (1st degree relatives), *n*(%)	
Coronary artery disease	14 (45.2)
Cerebrovascular disease	8 (25.8)
Diabetes (type 1 or 2)	19 (61.3)
Hypertension	25 (80.6)
Malignancy	21 (67.7)
Obesity	10 (32.3)
Gestational hypertension	6 (19.4)
Preeclampsia	6 (19.4)
Nulliparous, *n*(%)	21 (67.7)
Gestational weight gain, kg, median (IQR) ^a^	13.4 (8.1, 18.4)
Inadequate ^e^, *n*(%)	8 (25.8)
Adequate ^e^, *n*(%)	8 (25.8)
Excessive ^e^, *n*(%)	14 (45.2)
Systolic blood pressure at delivery, mmHg, median (IQR)	135.0 (125.0, 152.0)
Diastolic blood pressure at delivery, mmHg, median (IQR)	85.0 (78.0, 91.0)
Mode of delivery, *n*(%)	
Spontaneous vaginal	6 (19.4)
Induced vaginal	13 (41.9)
Planned Cesarean section	3 (9.7)
Emergency Cesarean section	9 (29.0)
Gestational age at delivery, weeks, median (IQR)	38.0 (36.3, 39.4)
Preterm (<37 weeks), *n*(%)	8 (25.8)
Term (≥37 weeks), *n*(%)	23 (74.2)
Infant birthweight, g, median (IQR)	3029 (2335, 3497)

IQR, interquartile range; BMI, body mass index; missing data for ^a^ *n* = 1; ^b^ *n* = 2 participants; ^c^ participants reported consuming >0–3 alcoholic beverages per week; ^d^ ≥150 min/week moderate to vigorous physical activity; ^e^ defined by 2009 Institute of Medicine (IOM) total gestational weight gain recommendations for singleton pregnancy.

**Table 3 jcm-11-04185-t003:** Participant characteristics at 6-months postpartum.

Characteristic	Overall, *n* = 31
Weeks postpartum, median (IQR)	26.0 (26.0, 27.0)
Waist circumference, cm, median (IQR) ^a^	91.4 (83.8, 104.1)
Hip circumference, cm, median (IQR) ^a^	106.0 (96.5, 119.4)
Maternal BMI, kg/m^2^, median (IQR)	28.1 (24.4, 35.5)
Underweight (<18.5 kg/m^2^), *n*(%)	1 (3.2)
Healthy (18.5–24.9 kg/m^2^), *n*(%)	8 (25.8)
Overweight (25–29.9 kg/m^2^), *n*(%)	11 (35.5)
Obese (≥30 kg/m^2^), *n*(%)	11 (35.5)
Postpartum weight retention, kg, median (IQR)	0.4 (−1.7, 7.0)
≥150 min/week moderate to vigorous physical activity ^a^, *n*(%)	16 (53.3)
Breastfeeding (exclusive or combination feeding) ^a^, *n*(%)	18 (60.0)
HbA1C, %, median (IQR)	5.3 (5.1, 5.5)
<4.8%, *n*(%)	2 (6.5)
4.8–6.0%, *n*(%)	29 (93.5)
>6.0%, *n*(%)	0 (0)
LDL, mmol/L, median (IQR) ^a^	3.0 (4.2, 5.6)
<2.6, *n*(%)	11 (36.7)
≥2.6, *n*(%)	19 (63.3)
HDL, mmol/L, median (IQR)	1.56 (1.2, 1.9)
<1.3, *n*(%)	9 (29.0)
≥1.3, *n*(%)	22 (71.0)
Triglycerides, mmol/L, median (IQR)	0.98 (0.73, 1.6)
<1.7, *n*(%)	23 (74.2)
≥1.7, *n*(%)	8 (25.8)
High-sensitivity CRP, mg/L, median (IQR)	2.6 (1.0, 9.6)
≤10, *n*(%)	23 (74.2)
>10, *n*(%)	8 (25.8)
Albumin:Creatinine ratio, median (IQR) ^b^	1.95 (1.2, 3.9)
≤2.0, *n*(%)	14 (56.0)
>2.0, *n*(%)	11 (44.0)
Systolic blood pressure, mmHg, median (IQR)	117.0 (111.0, 126.0)
<120 (optimal), *n*(%)	19 (61.3)
120–139 (not optimal), *n*(%)	5 (16.1)
140–159 (elevated), *n*(%)	1 (3.2)
≥160 or taking an antihypertensive medication (major), *n*(%)	6 (19.4)
Diastolic blood pressure, mmHg, median (IQR)	78.0 (73.0, 82.0)
<80 (optimal), *n*(%)	16 (51.6)
80–89 (not optimal), *n*(%)	8 (25.8)
90–99 (elevated), *n*(%)	1 (3.2)
≥100 or taking an antihypertensive medication (major), *n*(%)	6 (19.4)
Fasting glucose, mmol/L, median (IQR) ^a^	4.7 (4.4, 5.0)
≤6.88 (optimal), *n*(%)	30 (100)
>6.88 or previous diagnosis of type 1 or 2 diabetes (major), *n*(%)	-
Total cholesterol, median (IQR)	5.1 (4.0, 5.6)
<4.65 (optimal), *n*(%)	12 (38.5)
4.65–5.15 (not optimal), *n*(%)	4 (12.9)
5.16–6.19 (elevated), *n*(%)	14 (45.2)
≥6.20 (major), *n*(%)	1 (3.2)
Smoking, *n*(%)	
No (optimal)	30 (96.8)
Yes (major)	1 (3.2)
Lifetime CVD risk score, *n*(%)	
<39% (low)	13 (41.9)
≥39% (high)	18 (58.1)

IQR, interquartile range; BMI, body mass index; missing lab data for ^a^ *n* = 1, ^b^ *n* = 6 participants.

## Data Availability

The data presented in this study are available on request from the corresponding author. The data are not publicly available due to their containing information that could compromise the privacy of research participants.

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
