# Peer review of "Maternal Immune Cell and Cytokine Profiles to Predict Cardiovascular Risk Six Months after Preeclampsia"

_jcm, 2022, doi:10.3390/jcm11144185_

Round 1

Reviewer 1 Report

We know that preeclampsia syndrome (PE) complicates 2% to 8% of all pregnancies and that it can be the initiator of disorders at the endothelial level, superimposing on pre-existing circulatory, metabolic, hemostatic and immunological abnormalities. We also know that in the short term, PE is associated with serious maternal and fetal complications, and there is increasing evidence that in the 15 years after a problem pregnancy, PE is associated with a 2- to 7-fold risk of cardiovascular disease. elderly.

Malia S.Q. Murphy, et al. present an elegant study evaluating in a series of women (31 participants) who developed PE during pregnancy and up to 6 months postpartum and, using the presence of traditional cardiovascular risk factors at 6 months postpartum, whether the profiles of Cytokines and immune cells after PE are helpful in distinguishing women at low and high risk of CVD. Women who developed PE were followed up for immune cell phenotyping, plasma cytokine quantification at delivery, 3 months, and 6 months postpartum. Cardiovascular disease risk was assessed at 6 months postpartum, and immune cell and cytokine profiles were compared between risk groups at each time point.

This is a study with an impeccable methodology, and with very interesting results, which provides very relevant information on the impacts of PE on the regulation of the immune system, and which persist after pregnancy when it is normal to expect signs and symptoms of PE are resolved.

We invite you to continue your line of research by expanding your analyzes to a longer term, given that PE should be considered as a risk factor in the early development of cardiovascular disease in women, so that clinical detection strategies can be proposed. and intervention, for which they may benefit from early postpartum interventions.

Reviewer 2 Report

Discussion must be improved about existing literature of cvd

Reviewer 3 Report

This is a prospective cohort study regarding long-term prediction of complications from preeclampsia.  

In the methods- the PC ratio and microalbumin creatinine ratio is a ratio therefore it does not have any units of mg/dL or any units at all.  

What was the lifetime CV risk score based on ?  Has this score been validated?  Provide citations. 

Was there any baseline assessment of CVD risk before pregnancy?

How do you know that the preE impacted the immune system and not the other way around?  Perhaps there is reverse causality.  Perhaps patients at elevated risk of preE have preexisting immune derangements which also cause CVD.  

How is your immunological profiling any more useful than the CV risk score you used?  
